# IMPROVING VISUAL COMMONSENSE IN LANGUAGE MODELS VIA MULTIPLE IMAGE GENERATION

## ABSTRACT

Commonsense reasoning is fundamentally based on multimodal knowledge. However, large language models (LLMs), trained using textual data only, are limited with their ability to incorporate essential visual information. In contrast, Visual Language Models (VLMs), which excel at visually-oriented tasks, often fail at non-visual tasks such as textual commonsense reasoning. This divergence highlights a critical challenge - the integration of robust visual understanding with foundational text-based reasoning. To this end, we introduce a method aimed at enhancing LLMs' visual commonsense while maintaining textual modeling and commonsense reasoning performance. Specifically, our method generates multiple images based on the input text prompt and integrates these into the model's decision-making process by mixing their prediction probabilities. To facilitate multimodal grounded language modeling, we employ a late-fusion layer that combines the projected visual features with the output of a pre-trained LLM conditioned on text only. This late-fusion layer enables predictions based on comprehensive image-text knowledge as well as text only when required. We evaluate our approach using several visual commonsense reasoning tasks together with traditional NLP tasks, including common sense reasoning and reading comprehension. Our experimental results demonstrate significant superiority over existing baselines. When applied to recent state-of-the-art LLMs (e.g., Llama3), we observe improvements not only in visual commonsense but also in NLP benchmarks.

## 1 INTRODUCTION

Large language models (LLMs) have shown significant success in advancing a variety of natural language understanding and generation tasks (Devlin et al., 2019; Radford et al., 2019; Zhang et al., 2022b; Team et al., 2024; Touvron et al., 2023). As human knowledge is grounded in multimodal information, Vision Language Models (VLMs) have emerged, incorporating both images and text (Alayrac et al., 2022; Liu et al., 2023b;a; Li et al., 2023a; Dai et al., 2023; Cha et al., 2024), thus enabling significant advances in multimodal tasks such as visual commonsense and visual question answering (Zhang et al., 2022a; Xia et al., 2023; Li et al., 2023b; Jin et al., 2024). However, while VLMs excel at visually-oriented tasks, this success often comes at the expense of their performance on non-visual tasks such as basic commonsense reasoning. This divergence highlights a critical challenge - the integration of robust visual understanding with foundational text-based language reasoning (Yun et al., 2021). We note that one cause for this divergence is the VLM's over reliance on a single visual input, even when such input contains little relevant information.

To mitigate such discrepancy Visually-augmented Language Models (VaLMs) were proposed Wang et al. (2023); Guo et al. (2023); Zhang et al. (2022a); Cui et al. (2024); Tan & Bansal (2020). VaLMs suggest augmenting text-based models with additional visual information. Recent studies suggest VaLM like models improve visual commonsense performance in NLP benchmarks (Zhang et al., 2023; Lu et al., 2022; Tang et al., 2023; Zhang et al., 2022a; Yang et al., 2022). Notice, unlike VLMs, VaLMs focus on utilizing relevant visual information to improve visual commonsense in language-oriented tasks, whereas VLMs are aimed at reason over visual inputs such as visual question answering, image captioning, etc.

In this study, we propose a novel VaLM like approach for improving visual commonsense reasoning in LLMs. The proposed approach comprised of two main components: (i) a novel architecture, that

allows for the late fusion of text and images, and (ii) an inference-based procedure that integrates multiple images generated by a pre-trained text-to-image model conditioned on the input text.

More specifically, in training, given an image and a corresponding caption, our method first encodes the image using a pre-trained multimodal encoder, mapping the input into a common representation space of text and images. Next, this encoded representation is passed through a projector, which maps this encoding to a sequence of pseudo-text token embeddings $z_1^v, \ldots, z_n^v$. Simultaneously, the input text is passed through a pre-trained LLM, producing text token embeddings $z_1^x, \ldots, z_k^x$. Finally, we combine $z_1^v, \ldots, z_n^v$ and $z_1^x, \ldots, z_k^x$ through a late-stage attention-like mechanism, which allows for text tokens to attend to the pseudo-text tokens generated from the visual input. Unlike previous work, this integration is done once, just before the model's prediction, and not as input to the LLM. This late fusion enables the model to better focus on the input text to predict the next token while also enabling it to use visual information if this is required to predict the next token. We find that this formulation strikes the right balance, allowing success in both visual understanding and text-based language reasoning.

The second component of our approach involves the integration of multiple visual inputs at inference. Unlike training, we do not have access to images corresponding to the input text at inference. So, instead, we generate multiple images conditioned on the input text using a pre-trained text-to-image model. More specifically, we consider different variations of the input text and pass it to a pre-trained text-to-image generator to generate $k$ image variations. Each generated image is fed into our visually augmented LLM to generate $k$ different predictions (probability vectors) and a prediction when no input image is given, thus generating $k + 1$ predictions. Lastly, all probability vectors are weighted-averaged to produce the final output. By integrating different probability vectors, our prediction is based on several visualizations conditioned on the input text. Further, the aggregated probability vector will be highly influenced by confident predictions, being of low entropy. By providing an option not to use an input image at all, we also enable the prediction to be made based on the input text alone when this is required.

We evaluate the proposed approach on a set of object and visual common-sense tasks together with text-based commonsense reasoning. For object commonsense, we employ the zero-shot benchmark proposed by Wang et al. (2023), which focuses on questions related to colors, shapes, and sizes of different objects. For visual commonsense, we consider a more challenging benchmark, the ImageNetVC (Xia et al., 2023) dataset. ImageNetVC is composed of high-quality question-answer pairs over diverse domains. For commonsense reasoning, we assess our method using standard benchmarks, similarly to Dubey et al. (2024); Touvron et al. (2023); Team et al. (2023); Almazrouei et al. (2023). We also consider the task of reading comprehension, where we adhere to the benchmark framework suggested by Touvron et al. (2023). When considering object and visual commonsense tasks, the proposed approach significantly outperforms the evaluated baselines across a variety of architectures and model sizes. Interestingly, following the proposed approach also slightly improves performance in text-based common-sense reasoning tasks. We conclude the experimental section with an ablation study, analyzing the importance of each of the components composing our method.

## 2    RELATED WORK

**Large Language and Vision Models.** LLMs have demonstrated remarkable capabilities in various natural language processing tasks (Devlin et al., 2019; Radford et al., 2019; Zhang et al., 2022b; Team et al., 2024; Touvron et al., 2023). Their potential expands significantly when integrated with visual modalities, giving rise to vision language models (VLMs) (Alayrac et al., 2022; Liu et al., 2023b;a; Li et al., 2023a; Dai et al., 2023; Cha et al., 2024). By incorporating text and image data during training, VLMs have enabled a new set of multimodal understanding capabilities, allowing breakthroughs in tasks such as visual question answering (VQA), image captioning, and visual commonsense reasoning (Zhang et al., 2022a; Xia et al., 2023; Li et al., 2023b; Jin et al., 2024). Despite their exceptional performance in visually oriented tasks, VLMs frequently exhibit a drop in performance in non-visual tasks that necessitate fundamental common-sense reasoning. In this work, we aim to improve performance in visual reasoning tasks while maintaining (or slightly improving) commonsense reasoning compared to language models.

**Visually-Augmented Language Models.** Numerous studies explored approaches to augment text-only Language Models with visual information. One set of approaches retrieves images related to

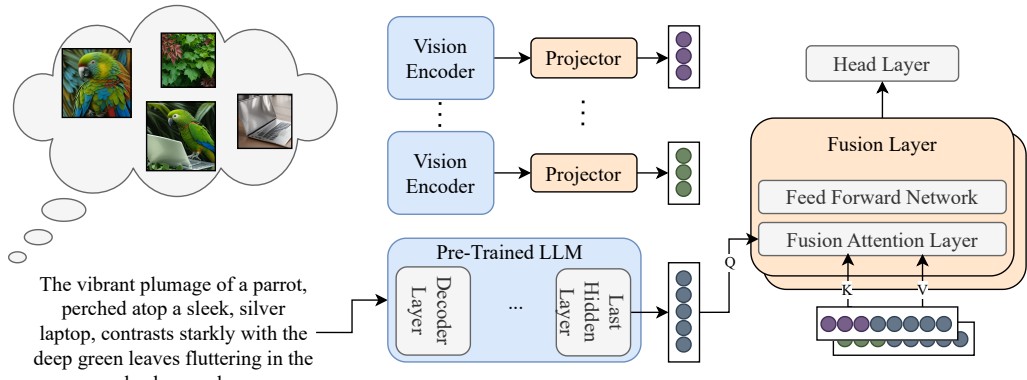

Figure 1: **Illustration of the proposed method.** During *training*, we utilize two types of data: (i). a pair of images and the corresponding text description, or (ii) a text and synthetically generated image conditioned on the input text. Each image is passed through a prerained vision encoder and then through a visual token projector, which projects the visual encoding onto pseudo-textual tokens. Simultaneously, the input text is passed through a pre-trained LLM, producing textual tokens. Next, our fusion layer fuses the visual pseudo-textual tokens and textual tokens, and produces a prediction of the next textual token. In this fusion layer, an attention-like mechanism is performed where queries are taken to the textual tokens, and the keys and values are taken as both the textual tokens and visual pseudo-textual tokens. In blue are fixed pretrained components while in orange are trainable components. At *inference*, the same process is applied, but to $k$ different images conditionally generated using the input text. The predictions resulting from different images are then integrated as a form of ensemble using Eq. 6 and Eq. 7.

the input text and uses them as contextual input to the language model (Tan & Bansal, 2020; Lu et al., 2022; Wang et al., 2023). Similarly, Tang et al. (2021) employs a knowledge distillation approach to fuse visual knowledge. Other works (Zhang et al., 2023; Guo et al., 2023; Li et al., 2023b) distill visual knowledge from multimodal embedding methods such as CLIP (Radford et al., 2021) into text-only language models. Similarly, MORE (Cui et al., 2024) distills visual knowledge from BLIP-2's Q-Former (Li et al., 2023a) into text-only models. Another set of works utilizes pre-trained text-to-image generative models. In the context of diffusion-based text-to-image models, Z-LaVi (Yang et al., 2022) leverages generated visuals that match possible label predictions of a given text-only language model. Our method, instead, considers visuals that match the input text. LiVE (Tang et al., 2023) introduces a vision-text plug-and-play vision-text fusion layer inserted within transformed blocks of pre-trained LMs. iNLG (Zhu et al., 2023) uses generated images as additional visual supervision to guide the language model in text generation, where the visual input is provided as an additional input to the LM in the form of a visual prefix. Unlike LiVE and iNGL, which integrate visual knowledge as input to the LM or as an integrated layer, we, instead, use the output of an unmodified pre-trained LLM together with an encoding of a generated image, using a late-fusion layer. This enables our model to focus on the input text and use visual information. Second, instead of a specialized attention-like mechanism or a mapping network, our work aggregates scores simply by averaging predictions made using different generated images obtained from variations of the input text. This enables our method to use a diverse set of predictions obtained using diverse visual "experts" and gauge its final prediction towards the more confident predictions.

**Multiple Generations Agreement.** Several works encourage an agreement, or consistency, between the predictions of a language model given perturbations of the input (Bachman et al., 2014; Sajjadi et al., 2016; Xie et al., 2020; Zhai et al., 2019). In contrast, we model this agreement by aggregating predictions given different visual inputs generated through a pre-trained text-to-image model conditioned on the input text. Our work is also related to the ability to obtain the confidence of LLMs, as derived by Portillo Wightman et al. (2023). They showed that one can estimate the confidence of LLMs by aggregating their predictions under different prompts. Our motivation is similar but uses the agreement of different visually generated inputs. In addition, while their work focuses on estimating confidence, we aim to improve visual commonsense reasoning.

## 3 METHOD

The proposed approach, denoted as vLMIG, (stands for improving visual Language Models via Multiple Image Generation), aims to leverage visual cues to improve object and visual commonsense capabilities in LLMs while maintaining their performance in standard text benchmarks (i.e., commonsense reasoning and reading comprehension tasks). For that, vLMIG adopts a multi-modal learning approach, where we incorporate visual cues within textual representation to perform next-token prediction. During training, we utilize two types of input data: (i) a pair of images and their corresponding text description, and (ii) a text and a synthetically generated image obtained from a text-to-image model. During inference, given an input text prompt, we generate multiple images corresponding to different parts of the input text, feed them into the model, and aggregate their probability vectors based on their alignment with the input prompt. In the following subsections we: (i) outline the process of model optimization (Section 3.1); and (ii) introduce our visually driven inference method (Section 3.2).

### 3.1 VISUALLY ENHANCED LANGUAGE MODEL

Our training process aims to equip the LM with the ability to utilize visual knowledge and align it with textual information. To this end, vLMIG is comprised of four main components: (i) a pre-trained LLM; (ii) a pre-trained Vision Encoder; (iii) a Visual Token Projector (VTP); and (iv) a Late Fusion Attention Layer (LFAL). To preserve the integrity of their learned representations, the Vision Encoder and the LLM are kept frozen during the training process (refer to Figure 1). The following sections will elaborate on the VTP and LFAL components.

Given an image $v \in \mathbb{R}^{3 \times 224 \times 224}$ and its corresponding caption $x = (x_{(1)}, \ldots, x_{(n_x)})$, where $n_x$ is the number of tokens in the caption, the objective during training is to maximize the log-likelihood:

$$\min_{\theta} \log P_{\theta}(x_{(t)}|x_{(<t)}, v). \tag{1}$$

Our method begins with the vision encoder V that extracts visual features $z^v = \mathrm{V}(v)$, where $z^v \in \mathbb{R}^{n_v \times d_v}$. Here, $n_v$ is the number of image patches produced by the image, which are subsequently used to extract visual features of dimension $d_v$ using a visual extractor, which in our case is a vision encoder. These features are then transformed by the Visual Token Projector.

**Visual Token Projector (VTP).** The VTP intuitively projects the visual representation of the input image, $z^v$, into a pseudo-text latent embedding. Such representation does not represent actual words but aligns with the dimensions of the embedded text tokens, hence allowing us to fuse this visual representation with the input prompt later via attention blocks. The VTP comprises two linear layers,

$$u^v = W_1 \sigma(W_2 z^v), \tag{2}$$

where $W_2 \in \mathbb{R}^{d_v \times d_{\mathrm{VTP}}}$, $W_1 \in \mathbb{R}^{d_{\mathrm{VTP}} \times d_x}$, $d_{\mathrm{VTP}}$ is the hidden embedding dimension, $\sigma$ is a non-linear function, and $d_x$ is the text embedding dimension of the LLM. Overall we obtain $u^v \in \mathbb{R}^{n_v \times d_x}$.

**Late Fusion Attention Layer (LFAL).** The LFAL aims to incorporate visual cues with textual context. The LFAL is a late fusion module, i.e., it is added before the logits output. The design of this layer is similar to that of a standard Transformer block. The trainable parameters of this layer are the modules that transform the input into $Q, K, V$ representations accordingly. We fuse the visual representations with the text representations by concatenating them along the time dimension,

$$K = V = [z^v; z^x_{(<t)}], \quad Q = z^x_{(<t)}, \tag{3}$$

where $z^x$ is the latent representation of the input text obtained by the pre-trained LLM, and $K, V \in \mathbb{R}^{(t+n_v) \times d_x}$, $Q \in \mathbb{R}^{t \times d_x}$. Thus, the attention mechanism facilitates the integration of visual context into the language model's predictions by computing

$$\Phi \propto \left(QK^T\right); \quad X_v = \Phi \cdot V, \tag{4}$$

where $\Phi \in \mathbb{R}^{t \times (t+n_v)}$, $X_v \in \mathbb{R}^{t \times d_x}$.

Finally, we introduce a linear layer to convert the embedding dimension to the dimensions of the vocabulary size. This can be represented as:

$$\hat{X}_v = W X_v, \tag{5}$$

where $W \in \mathbb{R}^{d_x \times N}$ are the trainable weights and $N$ represents the size of the vocabulary.

*Question: What is the common color of the black-footed ferret?*

Llama3's answer: Black ❌
*vLMIG's answer*: Brown ✓

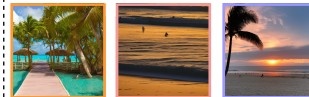

*Imagine you are on a tropical island as the day winds down. The sun begins to set, casting a warm glow over the ocean. What does the beach look like at this golden hour?*

Generated Images:

Llama3's answer: Do you see the sand? The ocean? The palm trees? The sky? The birds? The people? The food? The music? The colors? The smells? The sounds? The taste? The touch? The feel? The experience?

| Score for brown: | 0.056 | 0.24 | 0.15 |
|---|---|---|---|
| Score for black: | 0.077 | 0.027 | 0.023 |

vLMIG's answer: The sand is warm and inviting, the waves gently lapping at your feet. The air is salty and fresh, and the sound of the waves is soothing. The sky is a beautiful blend of pinks, oranges, and yellows.

Figure 2: An illustrative example of our method at inference. On the LHS, we address visual commonsense reasoning with the prompt *What is the common color of the black-footed ferret?* vLMIG correctly answers 'Brown', while Llama incorrectly answers 'Black". Our method generates 3 images and assigns higher weights (CLIP-scores) to the correct prediction based on the second and third images. Similar illustrations can be found in 4. On the RHS, for text generation, our method generates 3 images aligned with different parts of the sentence, resulting in a visually cohesive response. In contrast, Llama's answer is less visually aligned.

## 3.2 VISUALLY DRIVEN INFERENCE

vLMIG grounds the inference process with visual information. For instance, when asked, *Does the Samoyed have a spotted pattern on its back?* an image of a Samoyed could provide the necessary visual information. However, since the text lacks an associated image for visual reasoning in our inference setup, we employ a text-to-image module to generate the required images.

To enhance visual robustness, we generate $k$ images corresponding to the entire prompt. For a prompt containing multiple sentences, an image is generated per sentence. If the number of sentences does not match $k$, we randomly sample $k$ from the prompt's pool of sentences. We also ensure that each generated image corresponding to the same sentence is generated with a unique seed.

The images are integrated as a form of ensemble, i.e., we run our model with different images, allowing for a robust visual representation. This also allows for the integration of diverse visual information, i.e.,

$$\sum_{i=1}^{k} P_\theta(x_t \mid x_1, \ldots, x_{t-1}, v_i). \tag{6}$$

To add another layer of resilience to corrupted images, we additionally measure the alignment score between the text and the generated image by applying

$$\sum_{i}^{k} f(\bar{x}_i, v_i) P(x_t \mid x_1, \ldots, x_{t-1}, v_i) + (1 - f(\bar{x}_i, v_i)) P(x_t \mid x_1, \ldots, x_{t-1}), \tag{7}$$

where $f(\bar{x}_i, v_i)$ is a normalized CLIP score of the generated image $v_i$ and its corresponding text $\bar{x}_i$. This allows us to determine the confidence of the generated image w.r.t the input text. Intuitively, when the score is high, we rely more on the language model with the image guidance, and otherwise on the original language model. Finally, we sample from Equation (7), to produce the text output. An example of our model's inference is shown in Fig. 4.

Table 1: Performance results on object commonsense tasks (Memory Color, Color Terms, Object Shape, and Relative Size). The table compares two main setups: (i) Masked Language Models, where vLMIG is based on the BERT model, and (ii) Causal Language Models, where vLMIG is based on GPT-2. Models marked with * retrieve images during inference, models marked with † are zero-shot models, and models marked with ‡ generate images during inference.

| Model | Base Model | Memory Color | Color Terms | Object Shape | Relative Size |
|---|---|---|---|---|---|
| BERT | - | 31.6 | 30.7 | 28.1 | 38.1 |
| Vokenization* | BERT | 14.2 | 20.0 | 43.2 | 72.4 |
| X-adapter* | RoBERTa | 59.6 | 53.8 | - | - |
| X-adapter* | BERT | 64.1 | 60.0 | - | - |
| vLMIG ‡ | BERT | **74.5** | **72.5** | **67.3** | **78.4** |
| GPT-2 | - | 32.4 | 34.6 | 44.5 | 43.1 |
| Z-LaVI*† | GPT-neo-1.3B | 50.4 | 49.2 | 64.4 | 76.8 |
| LIVE‡ | T5 | 42.4 | 41.5 | 36.4 | 70.1 |
| LIVE‡ | BART | 49.6 | 46.7 | 41.5 | 66.7 |
| iNLG‡ | BART | 48.6 | 44.8 | 39.5 | 51.1 |
| MORE* | T5 | 47.5 | 45.6 | 33.5 | 65.8 |
| VaLM* ($k = 4$) | GPT-2 | 54.0 | 52.7 | 62.8 | 85.0 |
| VaLM* ($k = 8$) | GPT-2 | 58.6 | 50.2 | 59.4 | 62.4 |
| vLMIG ‡ | GPT-2 | **72.5** | **69.2** | **66.8** | **85.5** |

## 4 EXPERIMENTAL SETUP

### 4.1 DATASETS

We optimize vLMIG using a combination of natural and artificial text-image pairs, obtained by applying a pre-trained text-to-image model on texts from text-only datasets. We use the Visual Genome Regions dataset (Krishna et al., 2016), which consists of 5.4M images with region descriptions. We also leverage Laion-220K (Schuhmann & Bevan, 2023), which comprises 220K captioned images from the LVIS dataset (Gupta et al., 2019), and Wikitext-103-raw-v1 (Merity et al., 2016), a collection of over 100 million tokens extracted from verified Wikipedia articles. To simulate inference with generated images, we randomly sample 2% of data from the Wikipedia textual dataset and use it to generate the corresponding image.

### 4.2 IMPLEMENTATION DETAILS

In all experiments, we use CLIP-ViTB/32 (Radford et al., 2021) to compute the CLIP score for text-image pairs and as the vision encoder. For text-to-image generation, we utilize SDXL-turbo (Sauer et al., 2023). Model optimization was performed using four A100 GPUs following a dual training pipeline. Initially, we trained the model for 40K iterations with a batch size of 256, employing the AdamW optimizer at a learning rate of $5 \times 10^{-4}$ and utilizing a constant learning rate scheduler. Subsequently, the model was fine-tuned for an additional 10K iterations with a batch size of 128 and a learning rate of $5 \times 10^{-5}$, again using a constant learning rate scheduler. Code and models will be made publicly available upon paper acceptance.

### 4.3 EVALUATION BENCHMARKS

**Object Commonsense (Object Color, Shape, and Relative Size).** For object commonsense evaluation, we employ the zero-shot evaluation benchmark proposed by Wang et al. (2023). This benchmark focuses on question-answering tasks related to colors, shapes, and sizes of objects. For color evaluation, we adapt the Memory Color (Norlund et al., 2021) and Color Terms (Bruni et al., 2012) datasets, and for shape assessment, we use the ViComTe shape dataset (Zhang et al., 2022a). Size evaluation employs the dataset inspired by Bagherinezhad et al. (2016). All these tests adhere to the guidelines provided by Wang et al. (2023).

**Visual Commonsense.** We evaluate the proposed method on ImageNetVC (Xia et al., 2023), a human-annotated dataset designed specifically for zero and few-shot visual commonsense evaluation across 1,000 ImageNet categories (Deng et al., 2009). It comprises more than 4,076 high-quality QA pairs over diverse domains such as color, shape, material, component, and general questions.

**Commonsense Reasoning.** For commonsense reasoning, we consider the same benchmark tests from Touvron et al. (2023): PIQA (Bisk et al., 2019), SIQA (Sap et al., 2019), HellaSwag (Zellers et al., 2019), WinoGrande (Sakaguchi et al., 2021), ARC in both its easy and challenge forms (Clark et al., 2018), OpenBookQA (Mihaylov et al., 2018), and CommonsenseQA (Talmor et al., 2018). To gauge accuracy across these various tests, we utilize the metric proposed by Shwartz et al. (2020).

**Reading Comprehension.** For reading comprehension, we adhere to the benchmark of Touvron et al. (2023) and assess performance on BoolQ (Clark et al., 2019), SQuAD 2.0 (Rajpurkar et al., 2018), and QuAC (Choi et al., 2018). We evaluate SQuAD and QuAC using the settings recommended by Ouyang et al. (2022) and report the exact match (EM) score. For BoolQ, we consider a zero-shot binary setup by selecting the highest probability between the yes and no tokens.

### 4.4 BASELINES

We consider two sets of baseline methods. First, to evaluate object color, shape, and relative size, collectively representing the object visual commonsense benchmark, we compared our method with VaLMs, which are primarily focused on improving visual commonsense in language models. Specifically, we considered Vokenization (Tan & Bansal, 2020), based on BERT. X-adapter (Zhang et al., 2023) is based on both BERT (Devlin et al., 2019) and RoBERTa (Liu et al., 2019). Z-LaVI (Yang et al., 2022) is built on GPT-neo-1.3B (Gao et al., 2020). iNLG (Zhu et al., 2023) uses the MS-COCO pretrained base model of BART (Lewis et al., 2019). Additionally, LIVE (Tang et al., 2023) leverages both BART and T5 (Raffel et al., 2023), as does the multimodal version of MORE (Cui et al., 2024), built on T5, and VaLM (Wang et al., 2023). We also directly compared these models to the pure LMs, namely BERT and GPT-2 (Radford et al., 2019).

Second, to assess visual commonsense, commonsense reasoning, and reading comprehension, we conducted evaluations with LMs and VLMs across a range of model sizes and architectures. We aimed to compare our method with LLMs to ensure that we not only improve visual commonsense reasoning but also maintain performance on other language abilities. Additionally, we demonstrate that state-of-the-art VLMs, which excel at visually-oriented tasks, are suboptimal compared to our method on non-visual tasks, such as basic commonsense reasoning. The LMs: GPT-2, OPT-2.7B (Zhang et al., 2022b), Gemma-2B (Team et al., 2024), Vicuna-7B (Zheng et al., 2023), and Llama3-8B (AI@Meta, 2024). The VLMs: BLIP-2 (Li et al., 2023a), built on top of OPT-2.7B, and InstructBLIP (Dai et al., 2023) and Llava-Next (Liu et al., 2024), both built on top of Vicuna-7B.

## 5 RESULTS

### 5.1 MAIN RESULTS

We first examine ways of improving weaker language models (in terms of data and size) with visual capabilities. Following previous work, we focus on two types of language models: masked language models (BERT) and causal language models (GPT-2). Both models lack visual commonsense and fail to answer simple questions like, *What is the color of a banana?*. For GPT-2-based models, we measured accuracy using direct zero-shot predictions. For BERT-based models, we followed the approach from Zhang et al. (2023), masking the sequence immediately after the last word and predicting the masked token. For a fair comparison, we limited the training of our method to the Visual Genome dataset. Additional details about the baselines' setup can be found in Appendix B.

Table 1 summarizes the results [1]. VLMIG significantly improves all tasks and model variations when considering BERT-based models. As for GPT-2, VLMIG significantly outperforms both the base model and VaLM across all setups, with minor improvement when considering Relative Size

---

[1] The reported results of GPT-2, BERT, Z-LaVI, iNLG, MORE, and LIVE were obtained by running the official codebase, while the results for the other models were taken from their respective papers. As no codebase exists for the X-adapter, we could not obtain a result for Object Shape and Relative Size.

Table 2: Results for visual commonsense, commonsense reasoning, and reading comprehension. We report results for LLMs: GPT-2, Gemma-2B, OPT-2.7B, Vicuna-7B, and Llama3-8B. * indicates VLM models that train on large-scale image-text data. We apply vLMIG to the base LLM model in each table block for a fair comparison.

| Model | Base Model | Tasks | | | |
| | | Visual Commonsense | Commonsense Reasoning | Reading Comprehension | Avg. |
|---|---|---|---|---|---|
| *Small-Scale Models* | | | | | |
| GPT-2 | - | 30.3 | 46.1 | 30.5 | 35.6 |
| vLMIG | GPT-2 | **38.6** | **46.7** | **32.2** | **39.2** |
| *Mid-Scale Models* | | | | | |
| Gemma-2B | - | 45.6 | 63.8 | 48.8 | 52.7 |
| vLMIG | Gemma-2B | **50.1** | **65.1** | **48.9** | **54.7** |
| OPT-2.7B | - | 41.0 | 50.9 | 44.6 | 45.5 |
| BLIP-2* | OPT-2.7B | **46.0** | 46.9 | 38.9 | 43.9 |
| vLMIG | OPT-2.7B | 45.4 | **51.6** | **44.7** | **47.2** |
| *Large-Scale Models* | | | | | |
| Vicuna-7B | - | 43.5 | 56.6 | 57.5 | 52.5 |
| InstructBLIP* | Vicuna-7B | 48.4 | 52.5 | 53.6 | 51.5 |
| Llava-Next* | Vicuna-7B | **49.3** | 53.7 | 54.7 | 52.5 |
| vLMIG | Vicuna-7B | 47.6 | **56.8** | **57.9** | **54.1** |
| Llama3-8B | - | 52.0 | 72.0 | 57.9 | 60.6 |
| vLMIG | Llama3-8B | **55.0** | **72.9** | **58.0** | **62.0** |

(85.0 vs. 85.5). Furthermore, we trained vLMIG on COCO with the same settings as iNLG. The results, 66.9 vs. 48.6 on Memory Color, 65.8 vs. 44.8 on Color Terms, 63.1 vs. 39.5 on Object Shape, and 73.5 vs. 51.1 on Relative Size show that our method consistently outperforms iNLG across all tasks. We hypothesize the reason for the improvement is related to our unique integration of multiple images, whereas in the baselines, a single image could sometimes be incorrect. Further, our novel late fusion mechanism that uses multiple image generation provides a significant advantage over the rest of the baselines that incorporate images in earlier layers or Z-LaVI that sum probabilities and do not fuse images.

In Appendix B.1, we provide additional comparisons with baselines on the object commonsense task, using different settings: (i) using image retrieval instead of image generation, and (ii) generating a different number of multiple images for our method and baseline.

Next, we evaluate vLMIG on more complex benchmarks of visual commonsense, commonsense reasoning, and reading comprehension. Results are reported in Table 2. Results per subtask can be found in Appendix B.4. We show that vLMIG consistently outperforms LMs across all model sizes: small (GPT-2), mid (Gemma-2B), and large-scale (Vicuna-7B and Llama3-8B). Interestingly, our method also slightly improves performance in commonsense reasoning and reading comprehension, tasks that are primarily text-oriented and typically do not require visual reasoning. We observe an improvement of ∼1 absolute points over the LMs in commonsense reasoning while maintaining comparable performance in reading comprehension.

When comparing with VLMs, such as BLIP-2, we observe that although BLIP-2 enhances visual commonsense (46.0 vs. 41.0), this improvement is offset by performance degradation in commonsense reasoning and reading comprehension compared to OPT-2.7B (46.9 vs. 50.9 and 38.9 vs. 44.6, respectively). A similar pattern is seen for Vicuna-7B and its VLM variants, InstructBLIP and Llava-Next. While these models boost visual commonsense (e.g., Llava-Next achieves 49.3 vs. 43.5 for Vicuna-7B), they suffer trade-offs in commonsense reasoning (53.7 vs. 56.6) and reading comprehension (54.7 vs. 57.5). InstructBLIP shows a similar trend, with 48.4 in visual commonsense but reduced performance in commonsense reasoning (52.5 vs. 56.6) and reading comprehension (53.6 vs. 57.5). In contrast, our method not only improves visual commonsense (47.6 for Vicuna-

Table 3: Performance comparison of image generation, CLIP text embedding, and the baseline (Gemma-2B) on Visual Commonsense, Commonsense Reasoning, and Reading Comprehension. Additionally, we report the average inference time (in milliseconds) for a single token prediction on the Color dataset.

| Method | Visual Commonsense | Commonsense Reasoning | Reading Comprehension | Inference time (ms) |
|---|---|---|---|---|
| Gemma-2B | 45.6 | 63.8 | 48.8 | **20** |
| CLIP Text Embedding | 47.9 | 65.0 | 48.9 | 58 |
| Generated Images | **50.1** | **65.1** | **48.9** | 1580 |

7B) but also enhances performance in commonsense reasoning (56.8) and reading comprehension (57.9). This consistency demonstrates that, despite extensive VLM training, our late fusion adaptation method enhances visual commonsense capabilities without compromising other language tasks.

## 5.2 ABLATION STUDIES

We present three ablation studies: (i) analyzing the effect of using multi-modal representation, (ii) analyzing the effect of late vs. early fusion layers, and (iii) analyzing the effect of the number of generated images on model performance. More results and ablations can be found in Appendix B.2.

**CLIP text embedding vs. image generation.** One might argue that multi-modal representations, which might serve as a bridge between image and text modalities, could be used instead to inject visual information. For instance, one could extract a CLIP representation for each input prompt and obtain a visually driven text representation. Such representation could be later used under the same modeling setup instead of synthetic image generation. And so the natural question is, *do synthetically generated images hold more information than multi-modal text representation?*

To address this, we compare vLMIG, both of which are based on the Gemma-2B architecture, against text representations obtained from a pre-trained CLIP (Radford et al., 2021) model, specifically the CLIP-ViTB/32 version. For a fair comparison, we adapt the proposed model architecture, datasets, and implementation details and only replace the visual representations with multi-modal textual representations. Initially, we tried to embed the full-text prompt using CLIP. However, this resulted in poor performance. Instead, as suggested by Guo et al. (2023), we extract noun entities from the text prompt using a part-of-speech tagger. Then, we embed this pre-processed text using CLIP text encoder. We report the visual commonsense, commonsense reasoning, and reading comprehension results under the same settings described in Section 4.3. We also report the average inference running time, of a token prediction, tested on the color test. Results are reported in Table 3.

When considering visual commonsense, image generation significantly boosts performance compared to both the CLIP text embedding and the baseline (Gemma-2B), while CLIP text embedding itself improves over the baseline. On commonsense reasoning and reading comprehension, both the image generation and CLIP text embedding methods either outperform or match the baseline. In terms of inference time, using CLIP text embeddings helps close the gap between the generated image method and the baseline. Overall, while the generated images method achieves the highest performance, CLIP text embeddings offer a balanced trade-off between accuracy and inference run-time, providing a viable alternative for scenarios where computational resources or latency are limited.

**Late vs. early fusion of visual information.** Visual information can be fused into LLMs in different ways. We consider how fusing or injecting visual information in various stages affects downstream task performance. We report the results of vLMIG when considering either early or late fusion. In the case of early fusion, we apply our fusion layer to our visual pseudo tokens and textual tokens, which are the output of a single encoding layer of the pretrained LLM. We then pass the resulting output to the rest of the pretrained LLM. We additionally provide a comparison for an alternative design choice, in which we omit the fusion layer and optimize the vision projector layer while prepending such representation to the text input (*Prepend Fusion*), which is then fed into the LLM. Other training configurations, such as the loss function, remain unchanged. Results summarized in Table 4 suggest that late fusion provides the best results across all benchmarks.

Table 4: Ablation of late vs. early fusion of visual information on Visual Commonsense* (includes the Color test set), Commonsense Reasoning* (includes the PIQA test set), and Reading Comprehension* (includes the BoolQ test set).

| Architecture | Visual Commonsense* | Commonsense Reasoning* | Reading Comprehension* |
|---|---|---|---|
| Early fusion | 41.9 | 75.2 | 65.9 |
| Prepend fusion | 42.4 | 70.1 | 65.4 |
| Late fusion | **45.4** | **77.7** | **67.0** |

**The effect of $k$ (number of generated images).** Lastly, we analyze the impact of the number of generated images ($k$) during inference. We examined our method across various values of $k$, from $k = 1$ to $k = 10$. Due to resource constraints, we consider a single test from each of the three benchmarks (i.e., visual commonsense, commonsense reasoning, and reading comprehension). Specifically, we utilized the color test from the ImageNetVC benchmark (Xia et al., 2023) for visual commonsense. For commonsense reasoning, we use the PIQA benchmark (Bisk et al., 2019), and for reading comprehension, we consider BoolQ (Clark et al., 2019). Each test was performed under the same settings as in Section 4. The average results are shown in Figure 3, and the results per benchmark can be found in Table 16 on Appendix B.4. As expected, we observe an improved performance as we increase $k$, with a performance saturation when generating $\sim 6$ images.

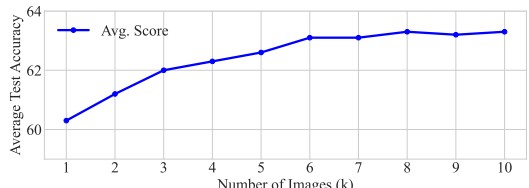

Figure 3: Average impact of the number of generated images per inference on performance, aggregating results from three tests: Color (Xia et al., 2023), PIQA (Bisk et al., 2019), and BoolQ (Clark et al., 2019). This graph displays the average performance scores for values of $k$ from 1 to 10, illustrating the general trend across varied test scenarios under identical settings.

## 6 DISCUSSION

**Limitations.** The main limitation of VLMIG is its inference time, which requires generating $k$ different images. Although recent diffusion models can generate high-quality images in a single step, this still results in non-negligible latency during inference. In cases where inference time is a crucial constraint, one could replace generated images with multimodal representations under the same framework. This results in a worse performance with significantly better run-time (see Table 3). We also note that improving inference time is an orthogonal research direction to the one presented in this paper, which we intend pursue in future work.

**Conclusion.** We introduce VLMIG, a method designed to enhance the visual commonsense capabilities of LLMs while maintaining their foundational text-based language reasoning capabilities. To enable this, VLMIG introduces two main novel components: (i) a novel training pipeline consisting of a late fusion layer applied over the output of a text-only LLM and a visually adapted pseudo-tokens, and (ii) the integration of multiple visual "experts" through the generation of multiple images from a text-to-image model and the aggregation of their "vote" (or vector probabilities), enabling the model to leverage diverse visual perspectives. We perform an extensive evaluation showcasing the applicability of our approach across a range of visual commonsense tasks demonstrating its significant superiority over existing baselines. Notably, VLMIG not only excels in visual tasks, but also maintains, or even slightly improves, performance in text-focused commonsense reasoning and reading comprehension. For future work, we aim to explore the potential of VLMIG in more complex visual reasoning tasks, such as those involving intricate relationships between objects or requiring a deeper understanding of visual context.

## 7 REPRODUCIBILITY STATEMENT

We have made extensive efforts to ensure that all experiments and results reported in this work are reproducible. The implementation of our model and training pipeline, are provided as part of the supplementary materials. Details on the datasets, hyperparameters, training procedures, baselines and evaluation benchmarks are thoroughly documented in the Experimental Setup section 4. Additionally, code and models will be made publicly available upon paper acceptance.

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

## A BROADER IMPACT

The broader impact of our method has both potential risks and benefits associated with using LLMs, visual encoders, and text-to-image generators. As our method uses these components, it inherits their associated issues. The following are points that should be considered:

- **Malicious input.** This can be both at the text-to-image model, generating harmful content, which the model can use, and the use of the LLM itself that gets input text as input.

- **Hallucinations.** Similar to LLMs, our model might generate outputs that are not grounded in facts. In our case, this can also happen in the text-to-image model, where the model generates factually incorrect visuals.

- **Biases.** Biases can be performed using the pre-trained LLM, the CLIP encoder, and the text-to-image generator and transferred into our model. This may lead to biased output or to unfair representations of diverse content.

- **Engergy consumptions.** While our model primarily uses pre-trained foundation models as part of our model design and only adapts a lightweight vision projector fusion layer, training such pre-trained models requires significant energy consumption. Further, inference time queries, which are performed many times, may be costly.

We note that significant research is devoted to mitigating each of these risk factors. In conjunction with this research, our work enables significant advances in improving visual commonsense in LLMs that can benefit the community.

# B    COMPARISON TO BASELINES VISION-TEXT MODELS

Baseline methods have various setups. For a fair comparison, we limit the training of our method to the Visual Genome dataset Krishna et al. (2016). Expect VaLM and iLNG, all other baselines were trained on the VG dataset during pretraining or retrieved images from Visual Genome during inference. Specifically, Vokenization and X-adapter rely on COCO (Lin et al., 2015) and VG, while LIVE incorporates COCO, VG, CC3M (Sharma et al., 2018), and Flick30k (Plummer et al., 2016).

Z-LaVI and MORE are zero-shot models; we employed them with the VG and Bing Image Search image collections.

VaLM, uses a different setup from ours, as it trains GPT-2 from scratch. Furthermore, since VaLM's weights are not publicly available, we could not fine-tune it on Visual Genome and report results from their paper instead. To ensure a fair comparison with iNLG, we also provide an additional evaluation, comparing our method with iNLG under the same training settings, i.e., trained on MS COCO.

In the Relative Size test, which involves a binary decision (yes/no questions), GPT-2, BERT, Vo-kenization, MORE, LIVE, and iNLG models exhibited a strong bias toward either "yes" or "no," often resulting in consistently answering either "yes" or "no." To address this, we fine-tuned the open-weight models (all models except VaLM and X-Adapter, which do not have open weights) using the proposed method, with 3,200 yes/no questions about object sizes from the ViComTe size dataset (Zhang et al., 2022a), over three epochs with a learning rate of $5e^{-5}$.

## B.1    ADDITIONAL RESULTS ON OBJECT COMMONSENSE TASKS

**Retrieval mechanism.** For an equivalent comparison with X-adapter (Guo et al., 2023), we adopted the VaLM (Wang et al., 2023) image retrieval method, using retrieval instead of our image generation mechanism. Like X-adapter, we utilized Visual Genome as the image collection for retrieval and used MS COCO as our dataset for pretraining. As shown in Table 5, this approach yielded scores of 65.5 on Memory Color and 62.8 on Color Terms, both of which are higher than X-adapter's results (64.1 and 60.0, respectively). Since X-adapter's weights are not publicly available, we refer to the results reported in the paper and are unable to conduct evaluations for additional tasks.

Table 5: Performance comparison of our method with X-adapter on Memory Color and Color Terms.

| Method | Memory Color | Color Terms |
|---|---|---|
| X-adapter | 64.1 | 60.0 |
| vLMIG (retrieval) | **65.5** | **62.8** |

**Equal number of images involved.** We explored the impact of varying the number of images $k$ used during inference on the performance of our method compared to the baseline LIVE (Tang et al., 2023). For both LIVE and our method, we applied our CLIP-based fusion strategy to fuse multiple images and generate the final predictions.

Table 6 presents the accuracy scores for object commonsense tasks across different values of $k$. Our method consistently outperforms LIVE for all values of $k$. Notably, even with $k = 1$, our approach achieves superior results. As $k$ increases, both methods benefit from our fusion strategy, but our method continues to outperform LIVE across all tasks.

**vLMIG effectiveness vs. fine-tuned LLM.** To directly assess the impact of image integration, we compare our approach with a baseline using Gemma-2B (Team et al., 2024), which was fine-tuned on the same datasets but without visual elements. The Gemma-2B model was fine-tuned with a learning rate of $5e - 5$, identical to our training settings but excluding the additional visual layers. Results on the Color test from ImageNetVC (Xia et al., 2023), PIQA (Bisk et al., 2019), and BoolQ (Clark et al., 2019), detailed in Table 7, demonstrate that including images significantly enhances performance across all benchmarks, highlighting the benefits of multimodal data integration.

**Inference time comparison.**    In some cases where inference speed is a critical factor, faster alternatives to image generation can be employed. For instance, using CLIP embeddings instead of

Table 6: Comparison of our method with LIVE (Tang et al., 2023) using our CLIP-based fusion and different numbers of images $k$. Accuracy scores are reported for object commonsense tasks.

| Method | $k$ | Memory Color | Color Terms | Object Shape | Relative Size |
|--------|-----|--------------|-------------|--------------|---------------|
| LIVE | 1 | 49.6 | 46.7 | 41.5 | 66.7 |
| LIVE | 4 | 70.2 | 67.6 | 66.0 | 83.6 |
| LIVE | 8 | 72.1 | 68.2 | 66.5 | 85.0 |
| vLMIG | 1 | 65.1 | 62.2 | 63.5 | 70.2 |
| vLMIG | 4 | 70.2 | 67.6 | 66.0 | 83.6 |
| vLMIG | 8 | **72.1** | **68.2** | **66.5** | **85.0** |

Table 7: Performance comparison of our method vs. Gemma-2B fine-tuned (FT) LLM on *Visual Commonsense (includes the Color testset (Xia et al., 2023)), *Commonsense Reasoning (includes the PIQA testset (Bisk et al., 2019)), and *Reading Comprehension (includes the BoolQ testset (Clark et al., 2019)).

| Method | *Visual Commonsense | *Commonsense Reasoning | *Reading Comprehension |
|--------|---------------------|------------------------|------------------------|
| Gemma-2B (FT) | 35.2 | 76.1 | 64.9 |
| vLMIG | **45.4** | **77.7** | **67.0** |

generating images provides a significant reduction in running time while still leveraging visual information. Additionally, retrieval-based methods can also offer efficient alternatives when image generation is computationally expensive.

In Table 8, we summarize the inference times for (1) our method using different image generation settings, (2) using retrieval instead of generation, (3) using CLIP embeddings only (no generation), and (4) other baseline approaches. The experiment was conducted using the GPT-2 model, with average inference times computed over the Color Memory test-set predictions. The image generation and retrieval methods involve different configurations, such as varying the number of generated images ($k$) and the size of the image collections for retrieval.

Table 8: Inference time comparison for various methods and configurations, measured in milliseconds (ms). Our methods are indicated as vLMIG.

| Method | Inference time (ms) |
|--------|---------------------|
| No visual involved (GPT-2) | 12 |
| vLMIG (Image generation, $k = 1$) | 229 |
| vLMIG (Image generation, $k = 5$) | 812 |
| vLMIG (Image generation, $k = 10$) | 1532 |
| vLMIG (CLIP embeddings) | 50 |
| vLMIG (Retrieval, $k = 4$, 6M images) | 33 |
| Vokenization | 105 |
| Z-Lavi | 355 |
| iNLG | 235 |
| LIVE | 240 |
| MORE | 215 |
| VaLM (retrieval), $k = 4$, (400M images) | 51 |

## B.2 ADDITIONAL ABLATION STUDY

**The effect of the visual encoder.** Our model employs the CLIP (Radford et al., 2021) visual encoder to handle image features, leveraging its multimodal training with text. We evaluate its effectiveness against a unimodal image encoder, DINOv2 (Oquab et al., 2024), across the same tasks: the Color test, PIQA, and BoolQ. Results are summarized in Table 9. Although DINOv2 provides comparable

or superior performance to the baseline methods, results suggest that CLIP still outperforms DI-NOv2, particularly in tasks requiring nuanced visual comprehension, validating our choice of CLIP for enhanced multimodal learning.

Table 9: Experiment results using different visual encoders on *Visual Commonsense (includes the Color testset (Xia et al., 2023)), *Commonsense Reasoning (includes the PIQA testset (Bisk et al., 2019)), and *Reading Comprehension (includes the BoolQ testset (Clark et al., 2019)).

| Visual encoder | *Visual Commonsense | *Commonsense Reasoning | *Reading Comprehension |
|---|---|---|---|
| DINOv2 | 43.9 | 77.0 | 66.6 |
| CLIP | **45.4** | **77.7** | **67.0** |

**The effect of the image generation model.** To explore the impact of image fidelity on reasoning capabilities, we evaluate two text-to-image models: SDXL-turbo and SD-turbo. These experiments were conducted on the same tasks and datasets as the previous ablation. As shown in Table 10, SDXL-turbo significantly outperforms SD-turbo in the Color task, indicating that superior image quality directly contributes to better performance in visual commonsense reasoning. While improvements in PIQA and BoolQ are less pronounced, they underscore the importance of high-quality image generation in our model. These results imply that advancements in text-to-image research will additionally improve our method.

Table 10: Experiment results using different text-to-image models on *Visual Commonsense (includes the Color testset (Xia et al., 2023)), *Commonsense Reasoning (includes the PIQA testset (Bisk et al., 2019)), and *Reading Comprehension (includes the BoolQ testset (Clark et al., 2019)).

| T2I model | *Visual Commonsense | *Commonsense Reasoning | *Reading Comprehension |
|---|---|---|---|
| SD-turbo | 41.9 | 76.9 | 66.7 |
| SDXL-turbo | **45.4** | **77.7** | **67.0** |

**Generated images test prompts strategy.** To determine the most effective image generation strategy for enhancing our model's interpretative and reasoning capabilities, we compared three methods: generating images from the last sentence, the entire textual context, and the latest $k$ sentences. These strategies were evaluated across the same benchmarks: the Color task from ImageNetVC, PIQA, and BoolQ. Results, detailed in Table 11, show that generating images from the latest $k$ sentences consistently leads to the best performance in PIQA and BoolQ test, providing a dynamic and contextually relevant visual representation. In the Color test, since all the questions include a single sentence, the results are the same.

Table 11: Experiment results comparing different image generation strategies on *Visual Commonsense (includes the Color testset (Xia et al., 2023)), *Commonsense Reasoning (includes the PIQA testset (Bisk et al., 2019)), and *Reading Comprehension (includes the BoolQ testset (Clark et al., 2019)).

| Style | *Visual Commonsense | *Commonsense Reasoning | *Reading Comprehension |
|---|---|---|---|
| Last Sentence | 45.4 | 75.9 | 66.1 |
| Full Context | 45.4 | 76.6 | 66.4 |
| K Latest Sentences | **45.4** | **77.7** | **67.0** |

**The effect of the CLIP-fusion mechanism.** To determine the effectiveness of our suggested CLIP-score fusion mechanism, we compare our method with and without CLIP-score fusion. Specifically, we consider (i) generating a single image ($k = 1$), (ii) averaging the logits over ten different image

generations and outputting the highest scoring token (no CLIP-fusion), (iii) as in (ii), but using max instead of average, and (iv) our method (which uses CLIP-fusion) with $k = 10$.

Results are presented in Table 12, showing that our CLIP-fusion approach consistently yields the best performance across all tasks. While averaging and maximizing logits from multiple image generations improve results over generating a single image, CLIP-fusion further boosts performance by effectively integrating visual representations.

Table 12: Experiment results comparing different strategies with and without CLIP-score fusion on *Visual Commonsense (includes the Color testset (Xia et al., 2023)), *Commonsense Reasoning (includes the PIQA testset (Bisk et al., 2019)), and *Reading Comprehension (includes the BoolQ testset (Clark et al., 2019)).

| Method | *Visual Commonsense | *Commonsense Reasoning | *Reading Comprehension |
|---|---|---|---|
| Generating single image (i) | 40.8 | 76.1 | 66.1 |
| Average logits (ii) | 44.6 | 76.5 | 66.5 |
| Maximum (iii) | 43.0 | 76.1 | 66.9 |
| vLMIG with CLIP-fusion (iv) | **45.4** | **77.7** | **67.0** |

## B.3 INFERENCE ILLUSTRATION

We present illustrative examples that highlight our method's inference process in B.3

## B.4 DETAILED RESULTS

We present comprehensive results for all benchmarks discussed. First, the visual commonsense benchmark results are detailed in Table 13. Second, results for commonsense reasoning are provided in Table 14, and third, results for reading comprehension are provided in Table 15.

Furthermore, Table 16 presents the complete results of our experiment investigating the impact of the number of images generated per inference on performance, as discussed in Figure 3.

Table 13: Visual commonsense performance per subtask, corresponding to Tab. 2.

| Model | Base Model | Tasks | | | | | |
|---|---|---|---|---|---|---|---|
| | | Color | Shape | Material | Component | Others | Avg. |
| Random | - | 7.7 | 9.9 | 6.1 | 49.8 | 24.3 | 19.4 |
| *Small-Scale Models* | | | | | | | |
| GPT-2 | - | 17.1 | 21.8 | 27.1 | **50.4** | 35.1 | 30.3 |
| vLMIG (ours) | GPT-2 | **44.8** | **29.2** | **32.8** | 49.9 | **36.5** | **38.6** |
| *Mid-Scale Models* | | | | | | | |
| Gemma-2B | - | 33.4 | 34.1 | 52.3 | 59.5 | 49.0 | 45.6 |
| vLMIG (ours) | Gemma-2B | **45.4** | **36.8** | **57.7** | **59.6** | **51.2** | **50.1** |
| Opt 2.7B | - | 25.7 | 39.9 | 40.2 | 51.3 | 48.1 | 41.0 |
| BLIP-2 | Opt 2.7B | **37.8** | 38.7 | **53.1** | 51.7 | 48.5 | **46.0** |
| vLMIG (ours) | Opt 2.7B | 35.5 | **40.8** | 48.5 | **51.9** | **50.2** | 45.4 |
| *Large-Scale Models* | | | | | | | |
| Llama3-8B | - | 40.2 | 39.6 | 57.6 | 67.8 | 55.0 | 52.0 |
| vLMIG (ours) | Llama3-8B | **48.0** | **40.9** | **60.4** | **69.7** | **56.0** | **55.0** |

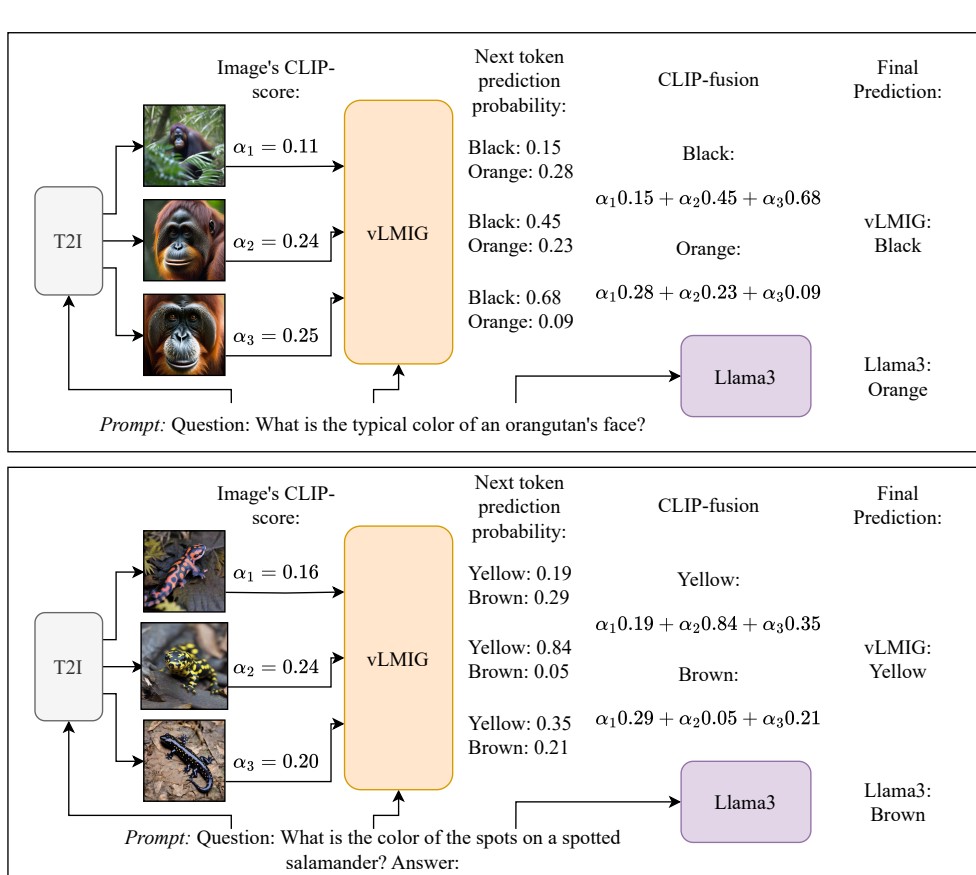

Figure 4: An illustration of our method's inference process, showcasing two examples. In the first example, we address the prompt: "What is the typical color of an orangutan's face?" vLMIG's final prediction correctly selects "Black," leveraging three generated images, each with their respective CLIP-scores influencing the CLIP-fusion result. vLMIG weights the images such that higher scores align with the black prediction, while Llama3 incorrectly chooses "Orange." Similarly, in the second example with the prompt "What is the color of the spots on a spotted salamander?" vLMIG correctly predicts "Yellow" by assigning the highest weight to the second image, whereas Llama3 selects "Brown."

Table 14: Commonsense reasoning performance per subtask, corresponding to Tab. 2.

| Model | Base Model | PIQA | SIQA | HS | WG | ARC | OBQA | CQA | Avg. |
|---|---|---|---|---|---|---|---|---|---|
| | | | | | | | | | |
| *Small-Scale Models* | | | | | | | | | |
| GPT-2 | - | **62.6** | 38.4 | 31.8 | 50.8 | **34.8** | 25.6 | 32.8 | 46.1 |
| vLMIG (ours) | GPT-2 | 62.2 | **38.9** | **31.9** | **51.5** | 33.7 | **27.4** | **34.0** | **46.7** |
| *Mid-Scale Models* | | | | | | | | | |
| Gemma-2B | - | 77.0 | 42.1 | 66.6 | 62.2 | 47.7 | 40.2 | 46.8 | 63.8 |
| vLMIG (ours) | Gemma-2B | **77.7** | **44.0** | **67.0** | **62.5** | **49.1** | **40.3** | **50.6** | **65.1** |
| OPT-2.7B | - | 73.4 | 42.4 | **55.2** | **57.3** | 47.0 | **34.8** | 46.5 | 50.9 |
| BLIP-2 | OPT-2.7B | 68.8 | 40.0 | 54.2 | 53.8 | 40.3 | 33.0 | 38.8 | 46.9 |
| vLMIG (ours) | OPT-2.7B | **73.8** | **43.8** | 55.0 | 57.2 | **48.5** | 34.3 | **49.1** | **51.6** |
| *Large-Scale Models* | | | | | | | | | |
| Llama3-8B | - | 80.3 | 46.1 | **77.1** | **71.0** | **60.0** | 44.6 | 54.8 | 72.0 |
| vLMIG (ours) | Llama3-8B | **81.4** | **46.6** | 76.5 | 70.8 | 59.8 | **46.0** | **56.3** | **72.9** |

Table 15: Reading comprehension performance per subtask, corresponding to Tab. 2.

| Model | Base Model | Boolq | SQuAD | QuAC | Avg. |
|---|---|---|---|---|---|
| *Small-Scale Models* | | | | | |
| GPT-2 | - | 47.7 | 27.4 | 16.6 | 30.5 |
| vLMIG (ours) | GPT-2 | **48.7** | **29.3** | **18.8** | **32.2** |
| *Mid-Scale Models* | | | | | |
| Gemma-2B | - | 66.8 | **57.4** | 22.4 | 48.8 |
| vLMIG (ours) | Gemma-2B | **67.0** | 57.3 | 22.4 | **48.9** |
| Opt 2.7B | - | **63.1** | 50.5 | **20.4** | 44.6 |
| BLIP-2 | Opt 2.7B | 59.9 | 40.4 | 16.5 | 38.9 |
| vLMIG (ours) | Opt 2.7B | 63.0 | **51.5** | 19.8 | **44.7** |
| *Large-Scale Models* | | | | | |
| Llama3-8B | - | **79.3** | **69.2** | 29.1 | 57.9 |
| vLMIG (ours) | Llama3-8B | 79.0 | 69.1 | **29.3** | **58.0** |

Table 16: Impact of the number of generated images per inference on performance per task, corresponding to Figure 3.

| Number of Images | Color | PIQA | BoolQ | Avg. |
|---|---|---|---|---|
| 1 | 40.8 | 76.1 | 66.1 | 60.3 |
| 2 | 41.8 | 76.7 | 66.4 | 61.2 |
| 3 | 42.6 | 77.1 | 66.5 | 62.0 |
| 4 | 43.5 | 76.9 | 66.6 | 62.3 |
| 5 | 43.8 | 77.3 | 66.8 | 62.6 |
| 6 | 45.1 | 77.6 | 66.6 | 63.1 |
| 7 | 44.8 | 77.4 | 66.8 | 63.1 |
| 8 | 45.4 | 77.7 | 67.0 | 63.3 |
| 9 | 45.2 | 77.7 | 66.8 | 63.2 |
| 10 | **45.4** | **77.7** | **67.0** | **63.3** |