# OpenReview forum: "Improving Visual Commonsense in Language Models via Multiple Image Generation"
_ICLR.cc/2025/Conference — Submitted to ICLR 2025_

### Official Review · Reviewer_Hoax · 2024-10-31

**Soundness:** 2
**Presentation:** 2
**Contribution:** 2
**Rating:** 5
**Confidence:** 2

**Summary:**

This paper propose a novel late-fusion method incorporating multiple images during training and inference to augment visual commonsense of LLMs. During inference, vLMIG leverage a Text2Image model to generate multiple images based on input text and use features of these image to enhance the KV in fusion layer upon the LLM.

**Strengths:**

1. Performance improvements  on object commonsense tasks are significant compared with baseline VaLMs.
2. The proposed method is simple to implement and ablation studies is comprehensive.

**Weaknesses:**

1. The reproducibility of results are questionable since the papers fails to clearly introduce the exact setting of the experimental results. For example, Sec. 4.1 states VG and Laion-220K are both used while #371 says only VG is used.
2. The comparison setting might be unfair. How to decide whether an incoming question needs image to augment or not? For questions that are irrelevant to visual knowledge, generated images can possibly harm the performance. Specifically, does the model generated images during evaluation on datasets like SQuAD.

**Questions:**

1. Are some paramaters frozen during training? The performance of vLMIG based on Llama3-8B is even improved on QuAC while only training on Visual Genome which exhibit limited text diversity.
2. Maintaining text understanding capability during training VLMs is studied comprehensively, such as MM1 and Deepseek-VL. I am curious if these VLMs can directly give good performance on tasks requiring visual commonsense knowledge while keeping a nice performance on textual tasks.

---

> ### Author Response · Authors · 2024-11-21
> **Response (part 1)**
>
> We thank the reviewer for the thoughtful and constructive feedback. Thank you for noting that our performance improvements on object commonsense tasks are significant compared with baseline VaLMs, that the proposed method is simple to implement, and that ablation studies are comprehensive. Below, we address the reviewer's concerns:
>
> 1. **Reproducibility information**. Thanks for noting this. For the results in Tab.1, we train only on the Visual Genome Dataset. However, for other results (Tab. 2 and ablations), we leverage the Visual Genome dataset together with Laion-220K and text-only data from Wikipedia (as described in Section 4.1, Datasets). This setting is fair both with respect to baselines in Tab. 1 and to baselines in Tab. 2. We agree that the writing doesn’t accurately describe this, and we will fully clarify this in the next revision. Beyond Sec. 4.1, the full experimental setup is provided in Appendix B, ensuring a fair comparison to baselines. We will be happy to clarify any further concerns the reviewer may have.
>
> 2.  **Comparison settings**. Thank you for noting this important concern. We have implemented three key strategies to address the dependency on the quality of generated images: (1). Our inference formulation, as detailed in Eq. 7, lines 259-269, weighs the contribution of each image by its alignment/relevance to the text using CLIP. Crucially, as noted in the RHS of Eq.7, when the CLIP score for all images is low (e.g., all noise or low res), most of the weight is provided to a text-only prediction head which does not consider the images in its prediction. This allows the model to gauge its prediction according to the relevance of input images. (2). We incorporated text-only data (e.g., from Wikipedia) during training, prompting the model to handle situations where relevant visual information is absent by focusing more on the text. (3). We generate multiple images (i.e., k=10) at inference, reducing the bias to a specific image through weighted averaging. For instance, as shown in Fig. 2, we generate three images. The RHS, for instance, uses all three images for the prediction. \
> To validate this, we tested our model under three different variations (using the number of images k as 10): (i). Replacing the generated images with images representing different prompts from the dataset, (ii). Using k-1 images representing different prompts from the dataset and a single generated image using the correct prompt, and (iii) Generating k images from correct prompts as default. The results, shown below, indicate that even when generated images are unrelated to the text context, our method performs comparably to the backbone on a visual commonsense task (corresponding to Tab. 2), with further improvements using one generated image and the best performance achieved with k generated images.
> | Approach                              | Accuracy |
> |---------------------------------------|----------|
> | Gemma-2B                              | 33.4     |
> | Images representing different prompts     | 33.0     |
> | k−1 images representing different prompts | 38.4     |
> | vLMIG (k generated images)            | 45.4     |
>
> Following the reviewer’s suggestion, we also consider an example of inference on SQuAD:
> When asking the question, “Which newspaper defined Southern California?” both Gemma-2B and vLMIG (Gemma-2B-based) respond with “The Los Angeles Times”, where all the generated images receive a CLIPScore below 0.2, indicating that none of them have a significant impact on the answer, equivalent to the backbone. When asked, “The Los Angeles Clippers are a team belonging to which sport?” The backbone provides a partial answer, “NBA”, which refers to the league, while vLMIG correctly answers “Basketball.” In this case, the generated images depict basketball scenarios, and 6 of them (out of 10) achieve a CLIPScore above 0.2, thus contributing more strongly to predicting the answer. Additional inference examples can be found in Fig. 4.

---

> ### Author Response · Authors · 2024-11-21
> **Response (part 2)**
>
> Questions:
>
> 1. **Frozen parameters**. Thanks for raising this point. Yes, we freeze the parameters of the backbone Pre-Trained LLM and vision encoder (see Fig. 1). This will be clarified.
>
> 2. **Visual-text balance**. Thanks for raising this interesting question. Deepseek-VL (https://arxiv.org/pdf/2403.05525) discusses the significant problem of language capabilities forgetting in LLMs (Fig. 4 of their paper) and suggests incorporating text-only data (specifically 70% multimodal data and 30% text-only data) to mitigate this problem. However, the presented results on the HellaSwag test set (commonsense reasoning) of ~0.685 accuracy is significantly worse than their backbone, Deepseek-LLM-7B, which achieved an accuracy of 0.754 (as reported in the Deepseek-LLM paper https://arxiv.org/pdf/2401.02954). In our evaluation, across all backbones, vLMIG achieves comparable results to the backbones on the HellaSwag test set. Further evaluation, provided below, supports this as well. It shows that, like other VLMs presented in Tab. 2, Deepseek-VL and Deepseek-LLM-7B perform better on visual commonsense tasks but worse on commonsense reasoning and reading comprehension.
> In the MM1 paper, they also discuss the problem of language capabilities forgetting in LLMs within VLMs (Fig. 5), and suggest the same solution as Deepseek-VL (incorporating text-only data). As we do not have access to the MM1 weights, we cannot present further evaluation but will be happy to present such an evaluation given a reference.
> | Method          | Visual Commonsense | Commonsense Reasoning | Reading Comprehension  | Avg. |
> |-----------------|--------------------|-----------------------|------------------------|------|
> | Deepseek-LLM-7B | 48.9               | **69.6**                  | **55.3**                   | **57.9** |
> | Deepseek-VL-7B  | **52.0**               | 65.5                  | 52.0                   | 56.5 |
>
> To summarize, we appreciate the thoughtful feedback from reviewer Hoax. We believe our clarifications and additional experiments address the points raised and further validate the contributions of our work. We will be happy to address any further concerns the reviewer may have.

---

> ### Comment · Reviewer_Hoax · 2024-11-23
>
> Thanks for the response. My concerns are partially resolved and I am willing to raise the score.
>
> I observe in Table 2. that the overall visual commonsense capability of models increases consistently with stronger LLM backbone. Is the performance improved with vLMIG which requires additional T2I model and inference cost more efficient than simply use a larger LLM?

---

> > ### Author Response · Authors · 2024-11-25
> > **Response**
> >
> > Thank you for the prompt reply and the additional question.
> >
> > vLMIG achieves better results more efficiently by bridging the gap between smaller and larger models in the visual commonsense task. This is achieved by leveraging the CLIP base model and SDXL-Turbo parameters, which together total ~2.5B parameters.
> > The tables below measure the average accuracy over the ImageNetVC (our Visual Commonsense benchmark) for different base models of different sizes. For the OPT base model, vLMIG based on OPT-2B performs comparably to the 66B version (Tab. A). The same holds for vLMIG based on Llama3-8B, which is comparable to Llama3-70B especially when using vLMIG with 3 LFALs that was suggested in the response “Larger scale results” for reviewer QLne. Since there is no mid-size Llama version between 8B and 70B, we include a comparison with Falcon (both 7B and 40B versions) (https://arxiv.org/pdf/2311.16867). Falcon’s 7B version performs close to Llama3-8B, while its 40B version, which uses ~4 times more parameters than our Llama3-8B version, achieves similar or worse performance, compared to us (Tab. B).
> > We additionally measure the inference time of vLMIG using Llama3-8B and Llama-70B models. To make sure the inference time is computed over an optimised inference process, we measure all run-time using the vLLM inference package (https://github.com/vllm-project/vllm). The inference time for vLIMG is 2425 ms (for generating 10 images), while the inference time for Llama 70B is 2802 ms. Hence, vLIMG (Llama-8B) is more efficient in terms of both number of params and inference time than the Llama-70B.
> >
> > Table A:
> > | Model | # of Parameters | Accuracy |
> > | --- | --- | --- |
> > | OPT-2.7B | 2.7B | 41.0 |
> > | vLMIG (OPT-2.7B) | 5.3B | 45.4 |
> > | OPT-66B | 66B | 45.7 |
> >
> > Table B:
> > | Model | # of Parameters | Accuracy |
> > | --- | --- | --- |
> > | Falcon-7B | 7B | 49.4 |
> > | Llama3-8B | 8B | 52.0 |
> > | vLMIG (Llama3-8B) | 10.7B | 55.0 |
> > | vLMIG (Llama3-8B) - 3 LFALs | 11.1B | 56.3 |
> > | Falcon-40B | 40B | 55.0 |
> > | Llama3-70B | 70B | 56.9 |
> >
> > This demonstrates that vLMIG effectively enhances smaller models to achieve competitive performance in visual commonsense reasoning while maintaining efficiency, avoiding the need to scale up to significantly larger LLMs.
> > We once again thank the reviewer for the prompt reply and for reconsidering the score. We believe the above addresses the reviewer’s further concern. We will be happy to address any further concerns the reviewer may have.

---

### Official Review · Reviewer_PmKL · 2024-11-03

**Soundness:** 3
**Presentation:** 3
**Contribution:** 3
**Rating:** 6
**Confidence:** 4

**Summary:**

This work proposed vLMIG, a novel visually-augmented language modeling framework for visual commonsense reasoning.

**Strengths:**

1. This work resolved a significant issue in current LLaVA-style MLLMs, in which the lightweight visual SFT always leads to significant catastrophic forgetting and performance degradation on text-only capabilities. VILA (Lin et al., 2023) resolved this via large-scale multimodal pre-training, while the efficient training pipeline for resolving this issue is not explored well. Concluding from the results in Table-2, LLaVA-Next also gets a 2.9 scores degradation compared with its LLM backbone vicuna-7b, while the vLMIG achieved a slightly +0.2 score improvement.

**Weaknesses:**

1. This architecture design only slightly differs from Flamingo, in which Flamingo adopts the cross-attention and this work adopts joint self-attention following VaLM. An ablation study on adopting cross-attention layer design is also appreciated to investigate some potential in architectural design.

2. An ablation study on the number of LFALs can also strengthen the contribution of this work. Flamingo adds cross-attention to each layer, while vLMIG only adds one LFAL. If there is a scaling effect on the number of introduced LFALs, then the users can decide how many LFALs to add to achieve a great tradeoff between performance and efficiency.

3. A significant baseline of LLaVA-llama3-8b is missed. You can consider to try to eval the https://huggingface.co/lmms-lab/llama3-llava-next-8b model and add the results to Table 2.

4. The efficiency is still another concern due to the image generation at inference stage in Table 3. Maybe retrieval is a better way? Or some explorations on efficient image generation can be also added to this work.

**Questions:**

1. How many A100 hours are used for your training?

2. Why an out-of-date Wikitext-103 dataset is adopted as the training corpora, maybe ShreadPajama is another better choice? Or you adopt this dataset for the density of knowledge?

---

> ### Author Response · Authors · 2024-11-25
> **Response (part 1)**
>
> We thank the reviewer for the thoughtful and constructive feedback. We are grateful for the reviewer’s appreciation of the contribution of our work, particularly with regard to resolving a significant issue of current LLaVA-style MLLMs - that of catastrophic forgetting and performance degradation on text-only capabilities. Below, we address the reviewer's concerns:
>
> 1. **Architecture design.** Thanks for noting this important point. Flamingo performs a cross-attention fusion layer in **every** layer of the base model. Flamingo's decoder layer also includes both self-attention and cross-attention blocks. In contrast, we utilize a single late fusion layer, implemented as a joint self-attention mechanism. By using a single late fusion layer, we are significantly more parameter-efficient.
> Following the reviewer's suggestion, we replaced our self-attention layer with a cross-attention layer (when performing late fusion), considering Gemma-2B as the backbone and reporting the benchmark results as in Tab. 2. The results are as follows:
> | Approach             | Visual Commonsense | Commonsense Reasoning | Reading Comprehension |
> |----------------------|--------------------|-----------------------|-----------------------|
> | cross-attention      | 49.8               | 62.8                  | 45.2                  |
> | joint self-attention | **50.1**               | **65.1**                  | **48.9**                  |
>
> As can be seen, using cross-attention is comparable to using joint self-attention in visual commonsense, but worse on the other task, i.e. commonsense reasoning and reading comprehension. This suggests that cross-attention, can perform well in visual-related tasks but worse than our suggested joint self-attention in text-related tasks. We will add this ablation to our next revision.
>
> 2. **An ablation study on the number of LFALs.** Following the reviewer’s suggestion, we conducted an ablation study on the number of LFALs. Specifically, we consider a single LFAL, 3 LFALs (i.e. 3 joint self-attention layers but all performed in a late fusion stage), and a joint self-attention in the corresponding positions to the cross-attention layers in Flamingo - i.e. the base models’ 6th, 12th and 18th layers.
> We consider Gemma-2B as the backbone and report the benchmark results as in Tab. 2. Additionally, we report the GFLOPs required for inferring 128 tokens with an initial context of 32 tokens. The results are as follows:
> | Approach                                 | Total # of Layers | GFLOPS | Visual Commonsense | Commonsense Reasoning | Reading Comprehension |
> |------------------------------------------|-------------------|--------|--------------------|-----------------------|-----------------------|
> | Gemma-2B                                 | 18                | 59.19  | 45.6               | 63.8                  | 48.8                  |
> | Single LFAL                              | 19                | 60.03  | 50.1               | **65.1**                  | **48.9**                  |
> | Three LFALs                              | 21                | **61.73**  | 51.2               | 64.2                  | 48.2                  |
> | Joint self-attention in layers 6, 12, 18 | 21                | **61.73**  | **51.4**               | 63.1                  | 46.2                  |
>
> Thus, adding joint self-attention layers inside the LLM backbone improves visual commonsense but results in suboptimal performance in commonsense reasoning and reading comprehension while requiring more GFLOPs. Similarly, utilizing three LFALs increases the GFLOPs needed and enhances visual commonsense, though the results for commonsense reasoning and reading comprehension remain close to those of a single LFAL.
>
> 3. **LLaVA-llama3-8b baseline.** Thanks. Following the reviewer's suggestion, we conducted the comparison to LLaVA-llama3-8b. As can be seen in the table below (corresponding to Tab. 2), LLaVA-llama3-8B is marginally better in visual commonsense tasks but underperforms in commonsense reasoning and reading comprehension in comparison to vLMIG based on llama3-8B and the base model of llama3-8B. We note that while such a tradeoff occurs compared to LLaVA-llama-8b, we outperform Llama3-8B on all three benchmarks.
> | Approach          | Visual Commonsense | Commonsense Reasoning | Reading Comprehension |
> |-------------------|--------------------|-----------------------|-----------------------|
> | Llama3-8B         | 52.0               | 72.0                  | 57.9                  |
> | LLaVA-llama3-8B   | **55.6**               | 68.5                  | 54.8                  |
> | vLMIG (Llama3-8B) | 55.0               | **72.9**                  | **58.0**                  |

---

> > ### Author Response · Authors · 2024-11-25
> > **Response (part 2)**
> >
> > 4. **Efficiency.** Thanks, indeed, there is a tradeoff between efficiency and accuracy. In time-sensitive applications, we offer an alternative solution: using CLIP text embeddings as the visual context instead of generated images, as discussed in Section 5.2, “CLIP Text Embedding vs. Image Generation,” with explicit inference running times provided in Table 8. This alternative is worse than our full approach while being better than the base model. In terms of speed, it is comparable to the base model. As for retrieval, this performs worse (see comparison to VaLM and X-Adapter in Tab. 1).
> > Additionally, as shown in Fig. 6, choosing a smaller k (i.e., generating fewer images during inference, such as k=6), can significantly accelerate inference (Tab. 8) while maintaining results comparable to the optimal.
> > We also note that we currently use an efficient T2I model of SDXL-turbo, which performs generation in a single step. This already speeds up inference significantly. However, as the literature proposes faster and improved 2D generative models, our model will see benefits in both speed and quality (see our ablation on the effect of improved generation on performance in Tab. 10) .
> >
> > **How many A100 hours are used for your training?** For training vLMIG on Llama-3, we use ~192 A100 hours, ~90 hours for training vLMIG on Gemma-2B and OPT-2.7B, and ~50 hours for training vLMIG on GPT2. Full training times for our method based on all base models will be added in the final revision.
> >
> > **Training our method with RedPajama:** Thanks. Following the reviewer’s suggestion, we applied our method to the RedPajama corpus instead of Wikitext-103, using the base model of Gemma-2B. Specifically, we utilized a random subset of the English version of RedPajama (with the same number of tokens as used in Wikitext-103), ran the experiment using the same implementation details as in the original setup, and evaluated it under the settings described in Tab. 2.
> > As shown in the table below, utilizing RedPajama improves results across all benchmarks:
> > | Text corpus  | Visual Commonsense | Commonsense Reasoning | Reading Comprehension |
> > |--------------|--------------------|-----------------------|-----------------------|
> > | Wikidata-103 | 45.6               | 63.8                  | 48.8                  |
> > | RedPajama    | **45.9**               | **64.1**                  | **49.4**                  |
> >
> > Due to time constraints, we could not train on the full RedPajama dataset during the timeframe of the rebuttal, but we aim to do so in the next revision.
> >
> > To summarize, we appreciate the thoughtful feedback from reviewer PmKL. We believe our clarifications and additional experiments address the points raised and further validate our work's contributions. We will be happy to address any further concerns the reviewer may have.

---

> > > ### Comment · Reviewer_PmKL · 2024-11-25
> > >
> > > Thanks for the clarification and additional results. I will keep the positive score.

---

### Official Review · Reviewer_QLne · 2024-11-05

**Soundness:** 3
**Presentation:** 3
**Contribution:** 3
**Rating:** 8
**Confidence:** 4

**Summary:**

This paper proposes a new method for Visually-augmented Language Models (VaLM). The method proposes using fixed (frozen) pre-trained LLM and visual encoders and trainable visual projection and multimodal fusion layers. The pre-trained components never see inputs from the other modality. The fusion layer feeds into the output (softmax) layer. A second contribution of the paper is to use multiple generated images at inference with weighted-average vectors. The method shows empirical improvement in visual understanding tasks while showing neutral-positive behavior on pure text-understanding tasks.

[Update after author rebuttal] Given authors' clear rebuttals with additional results, I have revised my rating from 6 --> 8.

**Strengths:**

The main strengths of the paper are as follows:

* The empirical studies are conducted on variety of models (BERT, GPT-2, Llama-8b, Gemma-2B) which is commendable and also shows the broader applicability of the method.

* The empirical improvements on the datasets presented in the paper are significant, especially at smaller scale models (BERT, GPT-2, Gemma-2B).

* The proposal method is simple and easy to integrate in any existing LLMs.

**Weaknesses:**

The main weaknesses of the paper are:

* While the proposed method shows significant improvements in visual understanding tasks at small scale, the results at larger scale (namely Llama-8b in Table 2) show very marginal improvements over the baseline which suggests either (a) the late fusion network requires many more parameters (more # of transformer layers) as the base model becomes more capable, or (b) at larger base models, we need a more end-to-end approach where we feed concatenated text+image tokens at the input layer as opposed to the proposed late fusion approach.

* The visual understanding dataset used in Table 1 (object common sense) seems too simple to form a very informed opinion of the proposed method. The authors mention running this method on more complex datasets as future work but we may need to see some results during the discussion phase if possible.

**Questions:**

For the different vLMIG models in Table 1 & 2, how many layers are used in the fusion module?

---

> ### Author Response · Authors · 2024-11-24
> **Response (part 1)**
>
> We thank the reviewer for the thoughtful and constructive feedback. We are grateful for the reviewer’s appreciation of our method and empirical studies. We also thank the reviewer for noting the broad applicability of our method and that the empirical improvements on the datasets presented are significant. Below, we address the reviewer's concerns:
>
> **Ablations on Larger scale models (e.g., Llama-8b in Table 2)**. Thank you for noting this important point. To evaluate this, we consider the following variations, as suggested by the reviewer: Alternative (a). We propose a model where the late fusion layer comprises three transformer layers instead of one. Alternative (b). We replace the late fusion layer with an early fusion layer, where both text and image inputs are fed at the input, following the approach in Sec. 5.2. For a fair comparison, we consider two variants: one with one layer and a second with three layers, similar to our late fusion setup.
> We evaluate these approaches using the same datasets and implementation details as presented in Table 2. As shown in the table below, the late fusion approach performs significantly better than early fusion, consistent with the results in Section 5.2, row 477. We believe that, with early fusion, the text features are relatively raw, resulting in less heterogeneous representations, making it more difficult to fuse with the semantic visual CLIP embedding representations. In contrast, with the late fusion variants, the text features are encoded into a semantic space, enabling easier fusion with the semantic visual CLIP embedding representations.
> Furthermore, as the reviewer hypothesized, encoding the fused representations with three transformer layers instead of one further improves visual commonsense performance. Thank you for suggesting these important ablations.
> | Approach | Visual Commonsense | Commonsense Reasoning | Reading Comprehension |
> | --- | --- | --- | --- |
> | Llama3-8B | 52.0 | 72.0 | 57.9 |
> | Late fusion (a transformer layer) | 55.0 | 72.9 | 58.0 |
> | Late fusion (3 transformer layers) - Alternative (a) | 56.3 | 72.7 | 57.7 |
> | Early fusion (1 transformer layers) - Alternative (b) | 53.3 | 69.9 | 56.7 |
> | Early fusion (3 transformer layers) - Alternative (b) | 53.6 | 69.5 | 55.9 |
>
> In addition to the above, we compare our method with larger-scale models to show that, with a small increase in the number of parameters, vLMIG achieves significantly better performance, typically achieved with significantly more parameters in base LLM models
> The tables below measure the average accuracy over the ImageNetVC (our Visual Commonsense benchmark) for different base models of different sizes. For the OPT base model, vLMIG based on OPT-2B performs comparably to the 66B version (Tab. A). The same holds for vLMIG based on Llama3-8B, which is comparable to Llama3-70B, especially when using vLMIG with 3 LFALs. Since there is no mid-size Llama version between 8B and 70B, we include a comparison with Falcon (both 7B and 40B versions) (https://arxiv.org/pdf/2311.16867). Falcon’s 7B version performs close to Llama3-8B, while its 40B version, which uses ~4 times more parameters than our Llama3-8B version, achieves similar performance (Tab. B).
>
> Table A:
> | Model | # of Parameters | Accuracy |
> | --- | --- | --- |
> | OPT-2.7B | 2.7B | 41.0 |
> | vLMIG (OPT-2.7B) | 5.3B | 45.4 |
> | OPT-66B | 66B | 45.7 |
>
> Table B:
> | Model | # of Parameters | Accuracy |
> | --- | --- | --- |
> | Falcon-7B | 7B | 49.4 |
> | Llama3-8B | 8B | 52.0 |
> | vLMIG (Llama3-8B) | 10.7B | 55.0 |
> | vLMIG (Llama3-8B) - 3 LFALs | 11.1B | 56.3 |
> | Falcon-40B | 40B | 55.0 |
> | Llama3-70B | 70B | 56.9 |

---

> ### Author Response · Authors · 2024-11-24
> **Response (part 2)**
>
> **Visual commonsense benchmark seems too simple.** Thanks. We conducted an additional comparison on the VEC benchmark (https://arxiv.org/abs/2305.14057), which is a recently published benchmark for measuring object visual commonsense. Specifically, we present comparisons on additional tasks, such as Embodied Concepts, including objects' temperature, mass, hardness, and height.
> Each sample in the test set contains a sentence, a positive word (correct), and a negative word (incorrect). A response is correct if the model assigns lower perplexity to the sentence with the positive word compared to the one with the negative word. For example, in “Deep red fire is hotter than melted steel” (positive: hotter) versus “Deep red fire is colder than melted steel” (negative: colder), the model succeeds if it assigns lower perplexity to the first sentence.
> We report the accuracy for each dataset independently, evaluating vLMIG (Ours) based on GPT2, GPT-2, LiVE, iLNG, MORE, and Z-LaVI, the open weights baselines. As shown in the table below, vLMIG consistently outperforms all baselines. We will fully incorporate the results in our next revision.
> | Model | Base model | Height | Mass | Temperature | Hardness |
> | --- | --- | --- | --- | --- | --- |
> | GPT2 | - | 0.54 | 0.49 | 0.52 | 0.54 |
> | iLNG | BART | 0.58 | 0.49 | 0.51 | 0.56 |
> | LIVE | BART | 0.61 | 0.50 | 0.50 | 0.56 |
> | LIVE | T5 | 0.59 | 0.49 | 0.53 | 0.55 |
> | MORE | T5 | 0.52 | 0.50 | 0.52 | 0.55 |
> | Z-LaVI | GPT-neo-1.3B | 0.66 | 0.53 | 0.57 | 0.56 |
> | vLMIG | GPT2 | **0.71** | **0.66** | **0.60** | **0.58** |
>
> **For the different vLMIG models in Table 1 & 2, how many layers are used in the fusion module?**
> We use a single late fusion layer in both Tables 1 and 2. Please refer to our method’s section, L.200-215.
>
> To summarize, we appreciate the thoughtful feedback from reviewer QLne. We believe our clarifications and additional experiments address the points raised and further validate the contributions of our work. We will be happy to address any further concerns the reviewer may have.

---

### Official Review · Reviewer_2969 · 2024-11-12

**Soundness:** 3
**Presentation:** 3
**Contribution:** 2
**Rating:** 3
**Confidence:** 4

**Summary:**

The paper introduces a method called VLMIG, aimed at enhancing visual commonsense reasoning in large language models (LLMs) by incorporating multiple generated images into the inference process. The method is particularly innovative in its use of a late-fusion mechanism, allowing the language model to consider both text and visual information without compromising on text-based tasks. VLMIG integrates various image predictions generated by a pre-trained text-to-image model, weighting them to improve visual commonsense understanding. Experimental results demonstrate VLMIG’s effectiveness across several tasks, such as visual commonsense reasoning and traditional language benchmarks, outperforming current baselines and maintaining strong performance in non-visual tasks.

**Strengths:**

VLMIG’s late-fusion approach, which combines textual and visual data just before final prediction, allows it to perform well on both visual and textual commonsense tasks. This ensures that LLMs benefit from visual grounding while retaining their core language abilities. The model excels across multiple task types, including object commonsense tasks (e.g., color and shape recognition) and reading comprehension. This versatility makes it a well-rounded approach for both visual and language-based tasks. VLMIG consistently outperforms existing models, including VLMs and VaLMs, in visual commonsense benchmarks, showing that multiple image generation and aggregation is a promising direction for enhancing multimodal reasoning capabilities. The authors conduct thorough experiments, including ablation studies, to validate the effects of different components, such as multi-image generation and the late-fusion layer. This helps substantiate the robustness and adaptability of the proposed model.

**Weaknesses:**

1. The method relies heavily on the quality of generated images. Poor text-to-image generation could potentially harm performance, but this potential failure mode isn't thoroughly discussed. The effectiveness of VLMIG depends on the quality and relevance of the generated images. If the text-to-image model generates incorrect or low-quality images, this could lead to suboptimal or incorrect predictions, as the ensemble approach relies on the integration of various visual predictions
2. The approach requires generating and processing multiple images during inference, which likely increases computational costs and latency significantly. Although high-quality images can be generated quickly, this approach still results in considerable computational and time costs, especially when k is large. This could limit the method's practicality in time-sensitive applications
3. The paper doesn't adequately address how the approach scales with larger language models or longer text inputs, particularly given the computational requirements of processing multiple images.
4. There isn't sufficient analysis of the contribution of each component (VTP, LFAL, multiple image generation) to the overall performance.
5. Only test on VLMIG with GPT-2 backbone for table 1 may be really far away from current VLM community.

**Questions:**

As weakness

---

> ### Author Response · Authors · 2024-11-21
> **Response (part 1)**
>
> We thank the reviewer for the thoughtful and constructive feedback. We are grateful for the reviewer’s appreciation of our late-fusion approach and noting that its versatility makes it a well-rounded approach for visual and language-based tasks. We also appreciate the reviewer’s note that our model excels across multiple task types and that VLMIG consistently outperforms existing models, including VLMs and VaLMs, in visual commonsense benchmarks, showing that multiple image generation and aggregation is a promising direction for enhancing multimodal reasoning capabilities. Lastly, we appreciate the reviewer’s note that our model is well-validated across various settings and well-ablated. Below, we address the reviewer's concerns:
>
> 1. **The method relies on the quality of generated images**. We have implemented three key strategies to address the dependency on the quality of generated images: (1). Our inference formulation, as detailed in Eq. 7, lines 259-269, weighs the contribution of each image by its alignment/relevance to the text using CLIP. Crucially, as noted in the RHS of Eq.7, when the CLIP score for all images is low (e.g., all noise or low res), most of the weight is provided to a text-only prediction head which does not consider the images in its prediction. This allows the model to gauge its prediction according to the relevance of input images. (2). We incorporated text-only data (e.g., from Wikipedia) during training, prompting the model to handle situations where relevant visual information is absent by focusing more on the text. (3). We generate multiple images (i.e., k=10) at inference, reducing the bias to a specific image through weighted averaging. For instance, as shown in Fig. 2, we generate three images. The RHS, for instance, uses all three images for the prediction. \
> To validate this, we tested our model under three different variations (using the number of images k as 10): (i). Replacing the generated images with images representing different prompts from the dataset, (ii). Using k-1 images representing different prompts from the dataset and a single generated image using the correct prompt, and (iii) Generating k images from correct prompts as default. The results, shown below, indicate that even when generated images are unrelated to the text context, our method performs comparably to the backbone on a visual commonsense task (corresponding to Tab. 2), with further improvements using one generated image and the best performance achieved with k generated images.
> | Approach                              | Accuracy |
> |---------------------------------------|----------|
> | Gemma-2B                              | 33.4     |
> | Images representing different prompts     | 33.0     |
> | k−1 images representing different prompts | 38.4     |
> | vLMIG (k generated images)            | 45.4     |
>
> 2. **Computational costs**. Thanks, indeed, there is a tradeoff between efficiency and accuracy. In time-sensitive applications, we offer an alternative solution: using CLIP text embeddings as the visual context instead of generated images, as discussed in Section 5.2, “CLIP Text Embedding vs. Image Generation”. Explicit inference running times are provided in Table 8. This alternative is worse than our full approach while being better than the base model. In terms of speed, it is comparable to the base model.
>
> 3. **How the approach scales with larger language models or longer text inputs**. Unlike previous methods that use early fusion techniques, we employed a late fusion approach, which is an efficient way of fusing modalities as it only adds a single layer. Specifically, given a backbone with K attention layers, our late fusion layer, which utilizes the same attention backbone, has the same quadratic dependency on the textual input length as any of the K attention layers in the backbone. Thus, our total inference speed is the sum of our visual component (not dependent on text), the backbone (text-dependent), and our late fusion layer (text-dependent). Assuming a text context of size L, the base model’s dependency on the text is O(K * L^2), whereas ours is O((K+1) * L^2), which is only marginally higher. To validate this, we ran an experiment. Increasing the context length from a single sentence to 10 sentences (while keeping the number of generated images, k, at 10) in the backbone model increased the inference running time by a similar margin as vLMIG.
> | Approach                   | Inference running time (ms) |
> |----------------------------|-----------------------------|
> | Gemma-2B - single sentence | 20.82                       |
> | Gemma-2B - ten sentences   | 46.69                       |
> | vLMIG - single sentence    | 750.71                      |
> | vLMIG - ten sentences      | 779.34                      |

---

> ### Author Response · Authors · 2024-11-21
> **Response (part 2)**
>
> 4. **Analysis of the contribution**. We have included an ablation study in the paper addressing VTP (Tab. 4), LFAL (Tab. 9), and multiple image generation (Fig. 3, Tabs. 6, 11, and 16). We would gladly provide further analysis based on the reviewer's specific requests.
>
> 5. **Missing backbones**. The purpose of Tab. 1 is to compare our method with other state-of-the-art for Visually-augmented Language Models (VaLMs). The task of augmenting LLMs with visual input. These baselines report values (in their corresponding paper) on models of a similar scale, such as GPT-2. Since training all the baselines on larger models would require substantial computational resources, we consider models of comparable size to baselines, like GPT-2 and BERT, for a fair comparison. We would be glad to provide further comparisons given a reference. Additionally, to address performance on larger-scale models, in Tab. 2, we evaluate our approach against recent state-of-the-art LLM backbones.
>
> To summarize, we appreciate the thoughtful and constructive feedback from reviewer 2969. We believe our clarifications and additional experiments address the points raised and further validate the contributions of our work. We will be happy to address any further concerns the reviewer may have.

---

> ### Author Response · Authors · 2024-12-03
> **Response (part 1)**
>
> We would like to thank the reviewer for your follow-up comment. In the following paragraphs, we address your remaining concerns. Our response is split into two parts. In the first part, we provide a more fine-grained analysis of the inference run time under fixed model performance. In the second part, we provide several methods to improve inference run-time, which are left for future work.
>
> **Part A: inference run-time analysis under fixed model performance**
>
> We agree that vLMIG’s inference time is higher than the base model; however, the base model also performs worse than vLMIG on visual common-sense tasks. These additional inference costs are similar to other time-performance tradeoffs in LLMs. In cases where performance is of higher priority, inference costs will probably be larger (e.g., 8B LLM vs. a 70B LLM). Hence, to make a fair comparison w.r.t inference run time, we fix model visual commonsense accuracy on these tasks by using larger base models and calculate the overall overhead (both memory and inference run time).
>
> Table A presents the average accuracy over the ImageNetVC (the Visual Commonsense benchmark) for different base models of various sizes. vLMIG leverages the CLIP base model and SDXL-Turbo parameters, totaling approximately **2.5B** parameters. For the OPT base model, vLMIG based on **OPT-2B** performs comparably to the **66B** version.
>
> Table A:
> | Model | # of Parameters | Accuracy |
> | --- | --- | --- |
> | OPT-2.7B | 2.7B | 41.0 |
> | vLMIG (OPT-2.7B) | 5.3B | 45.4 |
> | OPT-66B | 66B | 45.7 |
>
> The same holds for vLMIG based on **Llama3-8B**, which is comparable to **Llama3-70B**, as shown in Table B, especially when using vLMIG with 3 LFALs suggested in the response **Larger scale results** for reviewer **QLne**. Since no mid-size Llama version exists between 8B and 70B, we include a comparison with Falcon (both **7B** and **40B** versions) (https://arxiv.org/pdf/2311.16867). Falcon’s 7B version performs close to Llama3-8B, while its 40B version, which uses ~4 times more parameters than our Llama3-8B version, achieves similar or worse performance than ours.
>
> Table B:
> | Model | # of Parameters | Accuracy |
> | --- | --- | --- |
> | Falcon-7B | 7B | 49.4 |
> | Llama3-8B | 8B | 52.0 |
> | vLMIG (Llama3-8B) | 10.7B | 55.0 |
> | vLMIG (Llama3-8B) - 3 LFALs | 11.1B | 56.3 |
> | Falcon-40B | 40B | 55.0 |
> | Llama3-70B | 70B | 56.9 |
>
> Next, we measure the inference time of vLMIG using Llama3-8B and Llama-70B models. To ensure the inference time is computed over an optimized inference process, we measure all run-time using the vLLM inference package (https://github.com/vllm-project/vllm). The inference time for vLIMG is **2425** ms (for generating ten images), while the inference time for Llama 70B is **2802** ms.
>
> To sum up, when visual common sense performance is of interest, vLIMG is more efficient than the LLM alternative both in terms of the number of parameters and inference run time.
> This demonstrates that vLMIG effectively enhances smaller models to achieve competitive performance in visual commonsense reasoning while maintaining the original model capabilities and model efficiency and avoiding the need to scale up to significantly larger LLMs. **Hence, we believe the proposed method is highly practical in cases where visual common sense tasks are of interest.**

---

> ### Author Response · Authors · 2024-12-03
> **Response (part 2)**
>
> **Part B: inference run-time and accuracy tradeoffs and potential future directions**
>
> Our research studies various ways of reducing inference time at the cost of accuracy performance and suggesting additional potential savings.
>
> 1. **Number of images:** As shown in Fig. 6, choosing a smaller $k$ (i.e., generating fewer images during inference, such as $k=6$), can significantly accelerate inference (Tab. 8) while maintaining results comparable to the optimal. We further note that running generation in parallel means that generating 6-10 images takes only slightly longer than a single image.
> 2. **Parallel computing:** The proposed method is based on late fusion layers; hence, text processing and image generation can be executed in parallel (until the fusing layer), making the technique highly parallelizable. In particular, assuming a text context of size $L$, the base model’s dependency on the text is $O(K \cdot L^2)$, where $K$ is the number of attention layers. As $L$ increases, the time taken for the base model surpasses that of the image generation stage or is comparable to it. As such, the overall time complexity, including the last fusion layer, becomes $O ((K+1) \cdot L^2)$.
> 3. **Dependence on the input length:** The additional $k$ image generation process has a fixed run-time complexity and is not dependent on the input length, $n$. In contrast, the LLM inference complexity is a function of the sequence length, i.e., the size of $n$. Hence, asymptotically, as $n$ grows, the difference between the methods will become smaller and smaller.
> 4. **Image resolution:** We use the CLIP base model as the image encoder, which accepts input images with a resolution of 224x224, while we generate images at a resolution of 512x512 (SDXL-turbo’s optimal resolution) and resize them before applying CLIP. Performing image generation at a lower resolution will significantly reduce the generation inference time. Initial runtime benchmarking we did resulted in a run time of 1938 ms for the Llama3-8B-based vLMIG compared to 1588 ms for the backbone, a negligible difference.
>
> In the community, an initial improvement in performance is commonly followed by efforts to optimize it. We leave the full potential of investigating the above mentioned direction as future research.
>
> Once again, we thank the reviewer for the prompt reply and for reconsidering the score. We believe the above response addresses the reviewer’s concern. We will be happy to address any further concerns the reviewer may have.

---

### Author Response · Authors · 2024-12-03
**Rebuttal Summary**

We sincerely thank the reviewers for their valuable feedback, which has greatly enriched our work.

The reviewers collectively recognized several key strengths in our approach:
- **Novelty**: Reviewers appreciated our novel late-fusion method and the use of multiple image generations to enhance visual - commonsense reasoning in language models (Reviewers 2969, QLne, PmKL, Hoax).
- **Strong performance improvements** on visual commonsense reasoning tasks (Reviewers 2969, QLne, Hoax).
- **Maintaining text understanding capabilities** while improving visual reasoning, thus addressing catastrophic forgetting in multimodal LLMs (Reviewer PmKL).
- **Versatility** of our model, across multiple task types (Reviewers 2969, QLne).
- **Simplicity and ease of integration** into existing LLMs (Reviewers QLne, Hoax).
- **Comprehensive experiments and ablation studies** validating our approach (Reviewers 2969, Hoax).

We have addressed the reviewers' concerns:
- **Efficiency:** Addressed computational costs and proposed alternatives for time-sensitive applications, enabling smaller models to achieve comparable performance to larger LLMs (Reviewers 2969, PmKL, Hoax).
- **Dependency on image quality:** Experiments show our method remains effective even with unrelated images (Reviewer 2969).
- **Scalability with larger models:** Conducted additional experiments demonstrating our method scales effectively with larger models (Reviewer QLne).
- **Additional benchmark:** Evaluated on the more complex VEC benchmark, outperforming all baselines (Reviewer QLne).
- **Architectural design and ablations:** Performed ablation studies confirming our design choices and demonstrating trade-offs between performance and efficiency (Reviewer PmKL).
- **Comparison with additional baselines:** Included comparisons with additional baselines, showing our method's superiority while maintaining or improving text-based performance (Reviewers PmKL, Hoax).
- **Training data:** Experiments with RedPajama corpus showed improved results (Reviewer PmKL).

Once again, we appreciate the reviewers' efforts, which have greatly strengthened our work. This has also led to increased scores (Reviewers QLne, Hoax). We believe we have thoroughly addressed all concerns and validated our contributions.

---

### Meta-Review · Area_Chair_iWmJ · 2024-12-23

**Metareview:**

The paper proposes vLMIG, a method for enhancing visual commonsense reasoning in language models by generating and incorporating multiple images during inference via a late fusion mechanism. The late-fusion approach allows LLMs to leverage visual information while maintaining text understanding capabilities, addressing the catastrophic forgetting issue common in multimodal models; The use of multiple generated images with weighted averaging improves robustness, which also shows promising improvements over baselines on visual commonsense reasoning tasks.

However, the main weakness is the computational overhead from generating multiple images during inference, though the authors demonstrate ways to mitigate this through parallel processing and using fewer images. In addition, while the method shows strong results on smaller models, the improvements become more marginal with larger backbones, suggesting potential scaling limitations. Most concerns have been addressed during the rebuttal, but some reviewers still remain unconvinced.

**Additional Comments On Reviewer Discussion:**

See above.

---

### Decision · Program_Chairs · 2025-01-22

Reject